# Evolutionary dynamics of cancer in response to targeted combination therapy

Ivana Bozic[1,2†], Johannes G Reiter[3†], Benjamin Allen[1,4†], Tibor Antal[5], Krishnendu Chatterjee[3], Preya Shah[6], Yo Sup Moon[6], Amin Yaqubie[7], Nicole Kelly[7], Dung T Le[8], Evan J Lipson[8], Paul B Chapman[7], Luis A Diaz Jr[9], Bert Vogelstein[9], Martin A Nowak[1,2,10]*

[1]Program for Evolutionary Dynamics, Harvard University, Cambridge, United States; [2]Department of Mathematics, Harvard University, Cambridge, United States; [3]Institute of Science and Technology Austria, Klosterneuburg, Austria; [4]Department of Mathematics, Emmanuel College, Boston, United States; [5]School of Mathematics, Edinburgh University, Edinburgh, United Kingdom; [6]Harvard College, Cambridge, United States; [7]Memorial Sloan-Kettering Cancer Center, New York, United States; [8]Department of Medical Oncology, Johns Hopkins University School of Medicine; The Sidney Kimmel Comprehensive Cancer Center at Johns Hopkins, Baltimore, United States; [9]Ludwig Center for Cancer Genetics and Therapeutics, Howard Hughes Medical Institute, Johns Hopkins Kimmel Cancer Center, Baltimore, United States; [10]Department of Organismic and Evolutionary Biology, Harvard University, Cambridge, United States

*For correspondence: martin_nowak@harvard.edu

†These authors contributed equally to this work

Competing interests: The authors declare that no competing interests exist.

**Abstract** In solid tumors, targeted treatments can lead to dramatic regressions, but responses are often short-lived because resistant cancer cells arise. The major strategy proposed for overcoming resistance is combination therapy. We present a mathematical model describing the evolutionary dynamics of lesions in response to treatment. We first studied 20 melanoma patients receiving vemurafenib. We then applied our model to an independent set of pancreatic, colorectal, and melanoma cancer patients with metastatic disease. We find that dual therapy results in long-term disease control for most patients, if there are no single mutations that cause cross-resistance to both drugs; in patients with large disease burden, triple therapy is needed. We also find that simultaneous therapy with two drugs is much more effective than sequential therapy. Our results provide realistic expectations for the efficacy of new drug combinations and inform the design of trials for new cancer therapeutics.

## Introduction

The current wave of excitement about targeted cancer therapy (*Sawyers, 2004*; *Sequist et al., 2008*; *Kwak et al., 2010*; *Chapman et al., 2011*; *Gonzalez-Angulo et al., 2011*) was initiated by the success of imatinib in the treatment of chronic myeloid leukemia (CML) (*Druker et al., 2006*; *Gambacorti-Passerini et al., 2011*). Four decades of research passed between the discovery of the Philadelphia chromosome and the first treatment to target an activated oncogene in a human cancer. Targeted therapies against many different types of cancer are now being developed at a fast pace. These include gefitinib and erlotinib for non-small-cell lung cancer patients with EGFR mutations (*Sequist et al., 2008*), panitumumab and cetuximab for metastatic colon cancer (*Amado et al., 2008*), vemurafenib for patients with melanomas harboring BRAF mutations (*Chapman et al., 2011*), and crizotinib for lung cancer patients with EML4-ALK translocations (*Kwak et al., 2010*). At present, dozens of other targeted cancer therapies have either been approved or are being evaluated in clinical trials.

**eLife digest** As medicine becomes increasingly personalized, more and more emphasis is being placed on the development of therapies that target specific cancer-causing mutations. But while many of these drugs are effective in the short term, and do extend patient lives, tumors tend to evolve resistance to them within a few months.

The key problem is that large tumors are genetically diverse. This means that for any given treatment, there is likely to be a small population of cells within the tumor that is resistant to the effects of the drug. When the drug is given to a patient, these cells will survive and multiply and this will lead ultimately to treatment failure. Given that a single drug is therefore highly unlikely to eradicate a tumor, combinations of two or more drugs may offer a higher chance of cure. This approach has been effective in the treatment of HIV as well as certain forms of leukemia.

Here, Bozic et al. present a mathematical model designed to predict the effects of combination targeted therapies on tumors, based on the data obtained from 20 melanoma (skin cancer) patients. Their model revealed that if even 1 of the 6.6 billion base pairs of DNA present in a human diploid cell has undergone a mutation that confers resistance to each of two drugs, treatment with those drugs will not lead to sustained improvement for the majority of patients. This confirms the need to develop drugs that target distinct pathways.

The model also reveals that combination therapy with two drugs given simultaneously is far more effective than sequential therapy where the drugs are used one after the other. Indeed, the model of Bozic et al. indicates that sequential treatment offers no chance of a cure, even when there are no cross-resistance mutations present, whereas combination therapy offers some hope of a cure, even in the presence of cross-resistance mutations.

By emphasizing the need to develop drugs that target distinct pathways, and to administer them in combination rather than sequentially, the study by Bozic et al. offers valuable advice for drug development and the design of clinical trials, as well as for clinical practice.

Although targeted agents have prolonged the lives of cancer patients, clinical responses are generally short-lived. In most patients with solid tumors, the cancer evolves to become resistant within a few months (*Amado et al., 2008*; *Sequist et al., 2008*; *Gerber and Minna, 2010*; *Chapman et al., 2011*). Understanding the evolutionary dynamics of resistance in targeted cancer treatment is crucial for progress in this area and has been the focus of experimental (*Engelman et al., 2007*; *Corcoran et al., 2010*; *Bivona et al., 2011*; *Diaz et al., 2012*; *Ellis et al., 2012*; *Misale et al., 2012*; *Straussman et al., 2012*; *Wilson et al., 2012*; *Khorashad et al., 2013*) and theoretical studies (*Dewanji et al., 2005*; *Komarova and Wodarz, 2005*; *Michor et al., 2005, 2006*; *Haeno et al., 2007*; *Dingli et al., 2008*; *Katouli and Komarova, 2010*; *Lenaerts et al., 2010*; *Beckman et al., 2012*; *Bozic et al., 2012*). One of the most important conclusions of these studies is that a small number of cells resistant to any targeted agent are always present in large solid tumors at the start of therapy and that these cells clonally expand once therapy is administered. Tumor recurrences are thus a fait accompli when single agents are delivered (*Diaz et al., 2012*).

How can one overcome the near-certainty of disease recurrence following therapy with such agents? Conceptually, there are two paths: treat tumors when they are very small, before a sufficient number of mutant cells conferring resistance have developed, or treat tumors with two or more drugs that target different pathways. In reality, the first option is usually not feasible, as clinicians have little or no control over the size of lesions in their patients at presentation. The second option, however, will become possible as more targeted agents are developed. The potential of combination therapy with targeted agents is buttressed by the success of conventional chemotherapeutic agents in leukemias and other cancers (*DeVita, 1975*) and of combination therapies for infectious diseases such as HIV (*Porter et al., 2003*). But the potential therapeutic utility of combination therapies targeting different pathways in solid tumors cannot be inferred from these prior studies, as the anatomic and evolutionary characteristics of solid tumors are far different from those of liquid tumors (leukemias) or infectious diseases. In this work, we have formulated a mathematical model to predict the effects of combined targeted therapies in realistic clinical scenarios and attempted to answer the question posed at the beginning of this paragraph.

## Results

Our model is based on a multitype branching process (see 'Materials and methods'). Similar mathematical modeling has successfully predicted the dynamics of acquired resistance, including the timing of treatment failure, in colorectal cancer patients treated with the EGFR inhibitor panitumumab (*Diaz et al., 2012*), and has led to specific recommendations for combination therapies to treat CML (*Komarova et al., 2009*; *Katouli and Komarova, 2010*). Our current work builds on these previous studies by using recent advances in the mathematical theory of branching processes (*Antal and Krapivsky, 2011*), which enable us to obtain results that are exact in the biologically relevant limit of many tumor cells and small mutation rate.

To obtain key parameters for our model, we have studied the dynamics of 68 index lesions in 20 melanoma patients receiving the BRAF inhibitor vemurafenib. The data from six patients that represented distinct patterns of responses are shown in *Figure 1*. Patients P1 and P2 achieved complete responses, and their lesions became undetectable. Patient P3 had stable disease, with tumors remaining approximately the same size throughout treatment. Patients P4 to P6 all had partial remissions, with some lesions shrinking and others unchanging or regrowing during treatment. As expected, the smallest lesions were the ones most likely to become undetectable when the agent was effective.

For 21 lesions in our vemurafenib dataset, two pretreatment measurements were available. Using these data, we calculate the average net growth rate of these lesions to be 0.01 per day, which is consistent with previous reports (*Friberg and Mattson, 1997*; *Eskelin et al., 2000*). The estimated average time between cell divisions in the absence of cell death in melanoma cells is 7 days (*Rew and Wilson, 2000*), implying a birth (cell division) rate of $b = 0.14$ per day. We set this as the typical birth rate, and additionally explore birth rates that correspond to a wide range of 1–14 days between cell divisions (*Supplementary file 3*). To achieve the observed net growth rate, we set the cell death rate to $d = b - 0.01$ (typical $d = 0.13$). We assume that these birth and death rates are valid for all cell types prior to treatment. For simplicity, we assume that these birth and death rates remain constant for all cell types prior to treatment, and neglect variations in the growth rate due to spatial and metabolic constraints in solid tumors (*Bozic et al., 2012*).

A given cancer therapy will reduce the birth rate and/or increase the death rate of tumor cells. A cell type is defined as sensitive if the treatment in question would cause its death rate to exceed its birth rate; otherwise, it is resistant. The key parameters describing a particular combination treatment are its effects on the birth and death rates of cells and the number of point mutations that have the potential to confer resistance. Consider a treatment with two drugs, 1 and 2. We denote by $n_1$ (respectively, $n_2$) the number of point mutations that have the potential to confer resistance to drug 1 alone (respectively, drug 2 alone). We denote by $n_{12}$ the number of point mutations that have the potential to confer resistance to both drug 1 and drug 2 (cross-resistance mutations). We assume that drugs in a combination treatment are given at concentrations tolerable by patients, and define the numbers of resistance mutations ($n_1$, $n_2$, $n_{12}$) relative to these concentrations (*Katouli and Komarova, 2010*).

A crucial quantity for the effects of combination therapy is the expected number, $X$, of resistant cells at the start of treatment in a lesion containing $M$ cells. From the dynamics of our branching process model (see *Supplementary file 1*), we obtain

$$X \approx M\left[ n_{12}\mu + \left( n_1 n_2 + \frac{n_{12}}{2}(n_1 + n_2 - n_{12}) \right)\mu^2 \right].$$

Here $\mu = \frac{u}{s}\log(Ms)$ (log denotes the natural logarithm), where $s = 1 - d/b$ is the survival probability of the branching process initiated with a single cell and $u$ is the point mutation rate, ~$10^{-9}$ for most cancers. As $\mu$ is small, this formula can be further simplified. If there is at least one possible mutation that could in principle confer resistance to both drugs, $n_{12} \geq 1$, then $X \approx M n_{12}\mu$. In this case, the expected number of cells resistant to both drugs is independent of the numbers of mutations, $n_1$ and $n_2$, that have the potential to confer resistance to each individual drug. Intuitively, this means that tumor cells are much more likely to become resistant to dual therapy through the occurrence of one mutation conferring resistance to both drugs simultaneously than through sequential mutations conferring resistance to each drug separately. If there is no mutation that could confer resistance to both

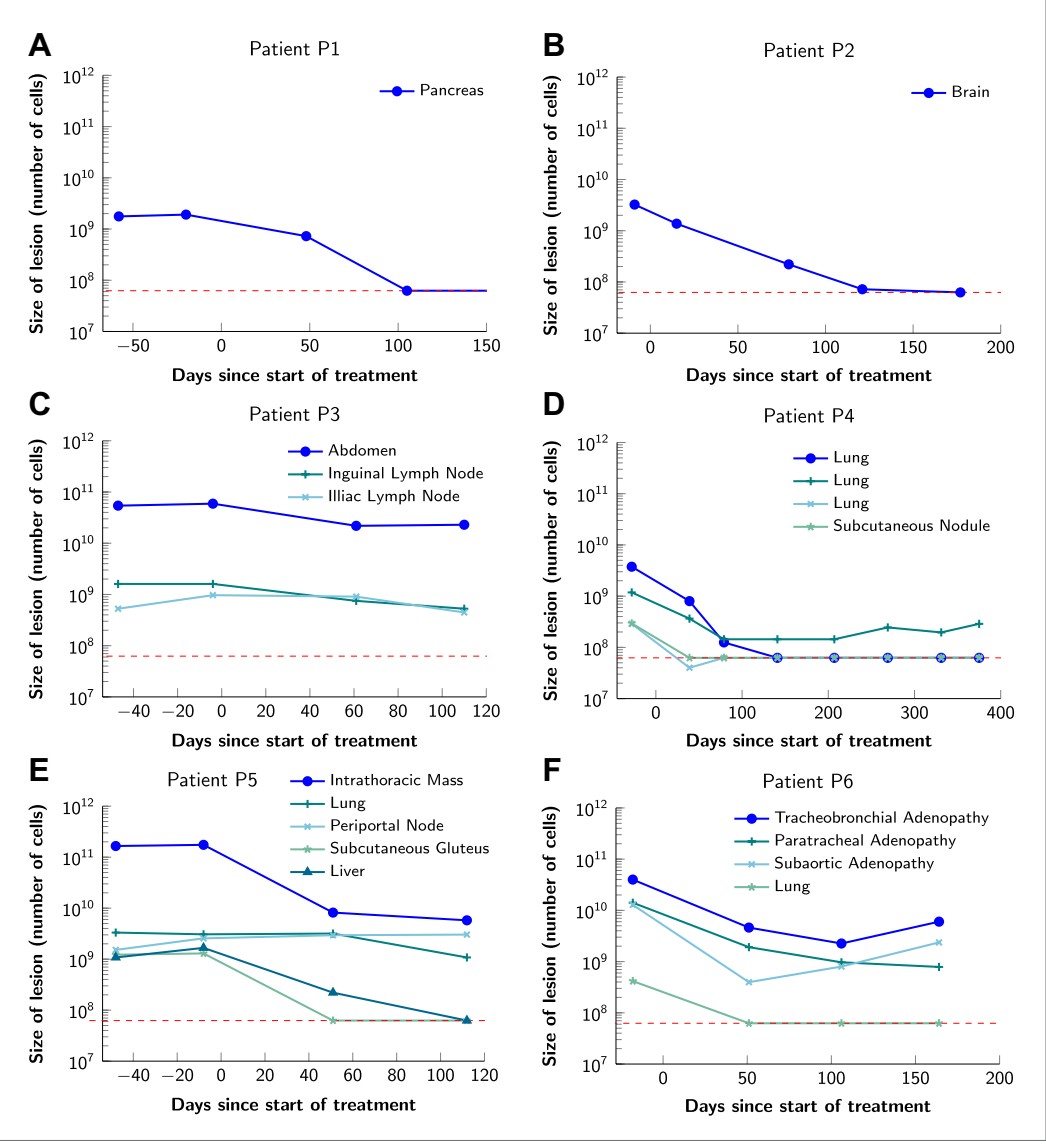

**Figure 1**. Variability in treatment response to monotherapy among six patients. Patients were treated with the BRAF inhibitor vemurafenib. Patients P1 and P2 achieved a complete response. Patient P3 had stable disease. Patients P4, P5, and P6 had partial responses. The minimal detection size (indicated by discontinuous red line) was assumed to be ≈63 × 10⁶ cells.

The following source data are available for figure 1:

**Source data 1**. Response to vemurafenib.

drugs simultaneously (no cross-resistance), then $n_{12} = 0$ and we obtain $X \approx M\, n_1 n_2\, \mu^2$. This quantity scales with the square of the point mutation rate, so the number of resistant cells in a tumor will be much smaller than for the case $n_{12} > 0$. In general, the expected number of cells resistant to combination therapy with $k$ drugs, with no cross-resistance, is $X \approx M\, n_1 n_2 \ldots n_k\, \mu^k$ (proof in ***Supplementary file 1***).

We emphasize, however, that resistance is the outcome of random mutation, division, and death events, and consequently may arise in one lesion but not in another, even if these lesions are otherwise identical. We therefore also obtain formulas for the probability that resistance to combination therapy is present at the time of detection. This probability can be computed as $p_{res} = 1 - p_1 p_2$. Here, $p_1$ is the probability that there is no resistance at detection that arose in a single mutational step, due to one of the $n_{12}$ possible cross-resistance mutations. $p_2$ is the probability that no such resistance arises in two

mutational steps. These probabilities can be expressed as follows (proofs in **Supplementary file 1**, Section 4):

$$p_1 = \exp\left( Mun_{12} \frac{\log(s)}{1-s} \right)$$

$$p_2 \approx \exp\left[ Mu^2(2n_1n_2 + n_{12}(n_1 + n_2)) \frac{\log(s)\log(Ms)}{s(1-s)} \right].$$

As above, $s = 1 - d/b$ is the survival probability of the branching process initiated with a single cell. The quantity $2n_1n_2 + n_{12}(n_1 + n_2)$ in the expression for $p_2$ represents the number of possible two-step mutational paths to dual resistance.

We turn now to the dynamics of the treatment response. Once treatment starts, sensitive cells decline, but resistant cells continue to grow. We assume that resistant cells maintain the pretreatment birth and death rates, $b$ and $d$, respectively, during treatment. To obtain estimates for the birth rate $b'$ and death rate $d'$ of sensitive cells during treatment, we calculate that the 68 lesions in our dataset declined at median rate $b' - d' = -0.03$ per day ($-0.01$ and $-0.07$ being 10th and 90th percentile, respectively). Thus, we set the typical death rate of sensitive cells during treatment to $d' = b' + 0.03$, and additionally explore cases when treatment is less ($d' = b' + 0.01$) or more effective ($d' = b' + 0.07$). As a default in our simulations, we suppose that treatment affects only the death rate ($b' = b$), but our mathematical analysis applies also to the case that treatment affects the birth rate.

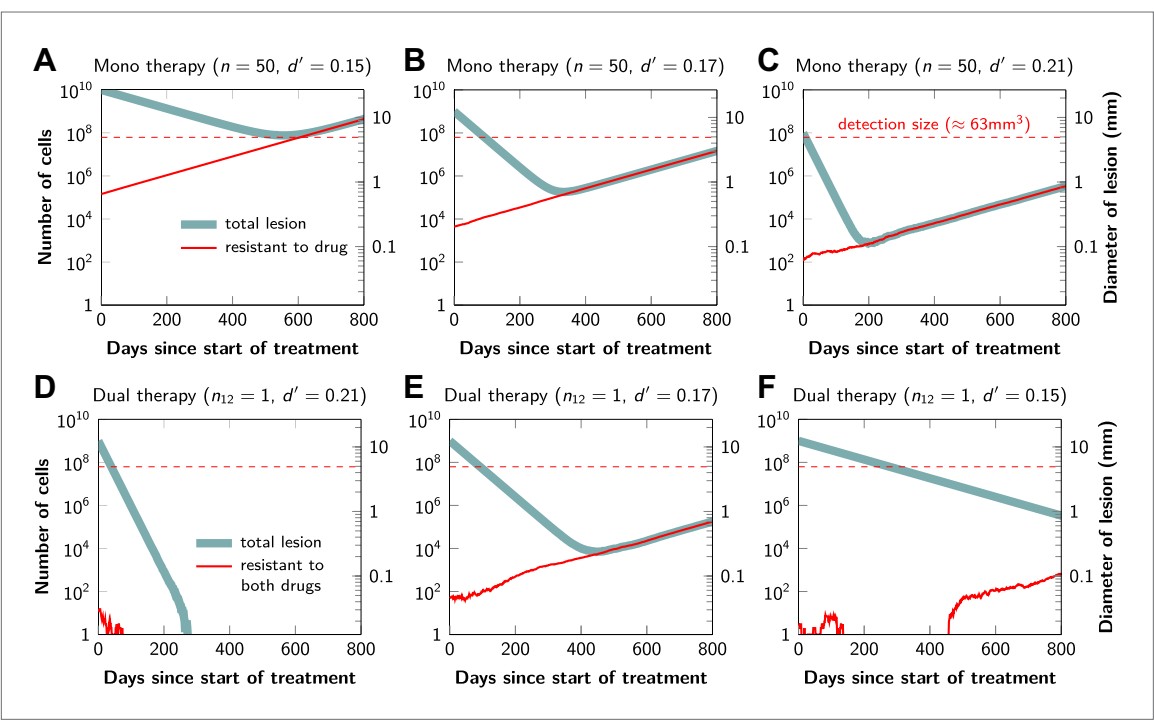

**Figure 2**. Tumor response to mono and dual therapy. The tumor grows exponentially until a certain detection size, $M$, is reached, at which point treatment is initiated. The number of point mutations that could in principle confer resistance to monotherapy is $n = 50$. For dual therapy, the number of point mutations that could confer resistance to drugs 1 and 2 separately is given by $n_1 = 50$ and $n_2 = 50$. The number of point mutations that could confer resistance to both drugs simultaneously is given by $n_{12}$. The point mutation rate was assumed to be $u = 10^{-9}$ and the rate of cell division $b = 0.14$ per day and is unaffected by treatment. The rate of cell death before treatment is $d = 0.13$ per day; it is increased to $d'$ for sensitive cells during treatment. (**A**)–(**C**) For clinically detectable sizes ($M = 10^{10}$, $10^9$, $10^8$, depending on the location of the tumors and the detection methods used), mono-therapy leads to a temporary shrinkage of the tumor but is always followed by tumor regrowth. (**D**) Due to stochastic fluctuations the few resistant cells present at the start of treatment go extinct and the lesion is eradicated. (**E**) Treatment leads to a temporary shrinkage of the tumor followed by regrowth. (**F**) The tumor decreases slowly in response to dual therapy, but resistant cells eventually evolve and cause treatment failure.

*Figure 2* shows computer simulations of single lesions in response to targeted therapies. Previous studies (*Engelman et al., 2007*; *Corcoran et al., 2010*; *Diaz et al., 2012*; *Ellis et al., 2012*; *Misale et al., 2012*; *Straussman et al., 2012*; *Wilson et al., 2012*) suggest that about 50 different mutations can confer resistance to a typical targeted therapeutic agent. Assuming that there are 50 or more potential resistance mutations, monotherapy will eventually fail in all lesions that can be detected by conventional imaging (*Figure 2A,B*) even when the death rate *d'* conferred by the therapy is far higher than usually observed in practice (*Figure 2C*). Small lesions, however, can decrease below the detection limit and appear to be eradicated for years before re-emerging (*Figure 2B,C*). This result is important, as it explains why tumors can recur after long periods of remission without the need to invoke processes involving cancer stem cells, angiogenesis, or immune escape (*Hensel et al., 2012*). Note that results similar to those obtained by simulation are observed in several of the individual lesions from actual patients graphed in *Figure 1*.

The results predicted to occur with dual therapy are shown in *Figure 2D–F*. Here, we also assume that there are 50 mutations that have the potential to confer resistance to either drug alone, but also that there is at least one mutation that can confer resistance to both drugs simultaneously. Intuitively, one might imagine that the existence of even a single cell resistant to both drugs at the start of therapy will automatically result in treatment failure. However, our results show that this is not necessarily true, and that the response depends on the size of the lesion, the number of cross-resistant cells, and the effects of the therapy on the balance between cell birth and cell death. Three examples illustrate these points. In *Figure 2D*, there is a small number of cells resistant to both drugs at the initiation of dual therapy, but these cells are lost by stochastic drift and the lesion is eradicated. In *Figure 2E*, there is a greater, but still relatively small number (~100), of cells resistant to both drugs. The lesion shrinks at first, but eventually progresses due to preexisting cross-resistance mutations within it. In the third lesion, the few cells resistant to both drugs at the initiation of therapy are lost to stochastic drift, but the cytolytic effects of the drug combination are less pronounced than in the other two cases (*d'* = 0.15 instead of 0.17 or 0.21). The relatively slow decrease in lesion size enables the generation of de novo resistance mutations during treatment and the lesion eventually recurred (*Figure 2F*).

In summary, treatment failure can be caused either by the preexistence of resistance to both drugs in a small number of tumor cells (*Figure 2E*) or the emergence of resistant cells during treatment (*Figure 2F*). Taking both of these possibilities into account, the probability, $p_{erad}$, that dual therapy eradicates a lesion containing *M* cells at the start of treatment is given by

$$p_{erad} = p_1^\uparrow p_1^\downarrow p_2^\uparrow p_2^\downarrow. \tag{1}$$

$p_1^\uparrow$ is the probability that no 1-step resistant lineage arises (and survives) prior to treatment. $p_1^\downarrow$ is the probability that no 1-step resistant lineage arises (and survives) during treatment. $p_2^\uparrow$ is the probability that no 2-step resistant lineage arises (and survives) prior to treatment. $p_2^\downarrow$ is the probability that no 2-step resistant lineage arises (and survives) during treatment. Here, 'steps' refers to the number of mutations (one or two) needed to achieve dual resistance, and 'lineage' refers to the descendants of a single cell that has achieved dual resistance via a particular mutational path. The therapy is successful if there is no resistant lineage arising in any of these four scenarios; since these are independent events, the overall success probability is obtained by multiplying the corresponding probabilities as shown in *equation (1)*. The probabilities that no 1-step resistant lineages arise before ($p_1^\uparrow$) or during treatment ($p_1^\downarrow$) and survive are given by *Komarova and Wodarz (2005)*

$$p_1^\uparrow = \exp(-Mun_{12})$$

and *Michor et al. (2006)*

$$p_1^\downarrow = \exp\left(Mun_{12}\frac{s}{s'}\right).$$

Here $s = 1 − d/b$ as above, and $s' = 1 − d'/b'$, where $b'$ and $d'$ are birth and death rates of cells sensitive to at least one drug during treatment (note that $s'<0$). The probabilities that no 2-step resistant lineages arise before ($p_2^\uparrow$) or during ($p_2^\downarrow$) treatment and survive can be calculated as:

$$p_2^\uparrow = \exp\left[ Mu^2 \frac{s'-s}{ss'}\left( n_1(n_2+n_{12}) \log\left(\frac{1}{sM}+u(n_2+n_{12})\frac{s'-s}{ss'}\right)\right.\right.$$

$$\left.\left.+n_2(n_1+n_{12})\log\left(\frac{1}{sM}+u(n_1+n_{12})\frac{s'-s}{ss'}\right)\right)\right]$$

and

$$p_2^\downarrow = \exp\left(-Mu^2\left(2n_1n_2+n_{12}(n_1+n_2)\right)\frac{s}{s'^2}\right).$$

The proofs of these results are provided in *Supplementary file 1*, Section 5. Excellent agreement between *equation (1)* and simulation results is shown in *Figure 3*.

Although modeling of single neoplastic lesions is the norm in theoretical studies, most patients with advanced cancers have multiple lesions and curing a patient requires eradication of all lesions. *Equation (1)* can be used to evaluate which combination treatments will be successful in typical patients with multiple metastatic lesions.

To determine the total extent of disease in typical patients who enroll for clinical trials, we quantified all radiographically detectable metastases in 22 such patients: 7 with pancreatic ductal adenocarcinomas, 11 with colorectal carcinomas, and 6 with melanomas—a different cohort than that depicted in *Figure 1*, in which only index lesions (those easiest to measure) were evaluated. The number of metastatic lesions in the 22 patients described in *Table 1* ranged from 1 to 30, and their total tumor burden ranged from $9 \times 10^8$ to $3 \times 10^{11}$ cells (see *Supplementary file 2*).

For each of these 22 patients, we used *equation (1)* to calculate the probability that monotherapy or dual therapy would eradicate all the patients' lesions. We find that monotherapy will fail in all 22 patients (*Table 1* and *Supplementary file 3*), as expected from the simulations in *Figure 2A–C* and

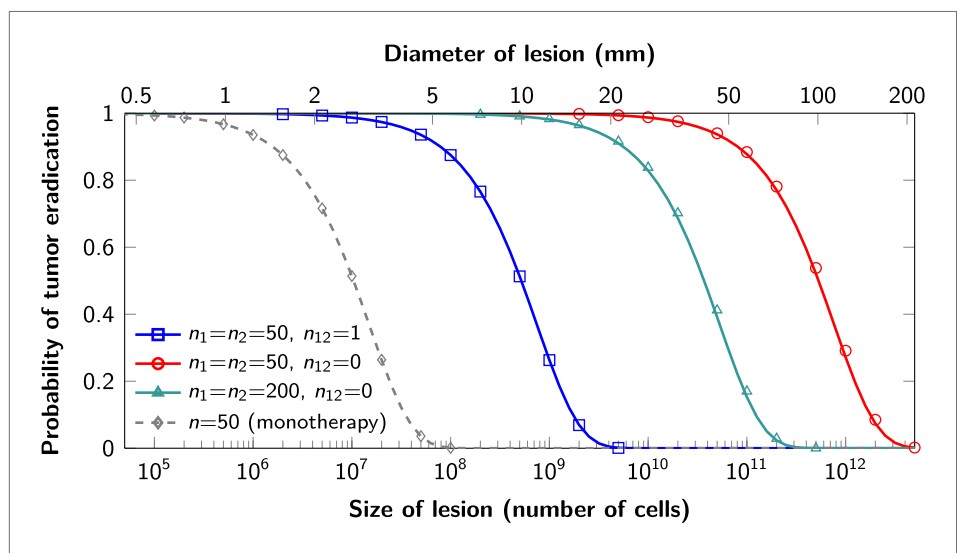

**Figure 3**. Probability of tumor eradication for two-drug combination therapy. A single mutation conferring cross-resistance to both drugs ($n_{12} = 1$) can prohibit any hope for a successful dual therapy. Solid curves show analytical results for dual therapy and dashed curve shows analytical results for a typical monotherapy, both are calculated using *equation (1)*. Markers (square, triangle, circle, diamond) indicate simulation results (averages of $10^6$ runs). Parameter values: birth rate $b$ = 0.14, death rate $d$ = 0.13, death rate for sensitive cells during treatment $d'$ = 0.17, point mutation rate $u = 10^{−9}$.

**Table 1.** Probability of treatment failure for combination therapy in patients

| Patient | Primary tumor type | Number of metastases | Total tumor burden (number of cells) | Probability of treatment failure | | |
| --- | --- | --- | --- | --- | --- | --- |
| | | | | Monotherapy | Dual therapy: $n_{12} = 1$ | Dual therapy: $n_{12} = 0$ |
| N1 | Pancreas | 18 | $2.6 \times 10^{11}$ | 1 | 1 | 0.283 |
| N2 | Colon | 25 | $2.3 \times 10^{11}$ | 1 | 1 | 0.26 |
| N3 | Melanoma | 26 | $1.7 \times 10^{11}$ | 1 | 1 | 0.203 |
| N4 | Melanoma | 30 | $1.4 \times 10^{11}$ | 1 | 1 | 0.172 |
| N5 | Colon | 21 | $1.0 \times 10^{11}$ | 1 | 1 | 0.128 |
| N6 | Melanoma | 8 | $9.8 \times 10^{10}$ | 1 | 1 | 0.12 |
| N7 | Colon | 25 | $9.1 \times 10^{10}$ | 1 | 1 | 0.112 |
| N8 | Pancreas | 8 | $7.4 \times 10^{10}$ | 1 | 1 | 0.092 |
| N9 | Pancreas | 23 | $6.4 \times 10^{10}$ | 1 | 1 | 0.08 |
| N10 | Pancreas | 5 | $5.5 \times 10^{10}$ | 1 | 1 | 0.069 |
| N11 | Colon | 14 | $5.4 \times 10^{10}$ | 1 | 1 | 0.068 |
| N12 | Rectal | 23 | $4.8 \times 10^{10}$ | 1 | 1 | 0.061 |
| N13 | Melanoma | 9 | $4.1 \times 10^{10}$ | 1 | 1 | 0.052 |
| N14 | Pancreas | 13 | $4.1 \times 10^{10}$ | 1 | 1 | 0.051 |
| N15 | Pancreas | 8 | $3.3 \times 10^{10}$ | 1 | 1 | 0.042 |
| N16 | Melanoma | 7 | $2.2 \times 10^{10}$ | 1 | 1 | 0.028 |
| N17 | Melanoma | 10 | $2.1 \times 10^{10}$ | 1 | 1 | 0.027 |
| N18 | Colon | 4 | $2.0 \times 10^{10}$ | 1 | 1 | 0.026 |
| N19 | Melanoma | 9 | $1.8 \times 10^{10}$ | 1 | 1 | 0.023 |
| N20 | Colon | 3 | $1.6 \times 10^{9}$ | 1 | 0.881 | 0.002 |
| N21 | Melanoma | 21 | $1.3 \times 10^{9}$ | 1 | 0.828 | 0.002 |
| N22 | Pancreas | 1 | $8.5 \times 10^{8}$ | 1 | 0.677 | 0.001 |

For monotherapy, we assume that 50 point mutations ($n = 50$) can in principle confer resistance to the drug. With dual therapy, we assume that 50 point mutations can in principle confer resistance to each drug individually ($n_1 = n_2 = 50$). Two scenarios are modeled: in the first, there is one mutation that can in principle confer resistance to both drugs (i.e., cross-resistance, $n_{12} = 1$). In the other case, there are no possible mutations that can confer resistance to both drugs ($n_{12} = 0$). Parameter values: birth rate, $b = 0.14$, death rate, $d = 0.13$, death rate for sensitive cells during treatment, $d' = 0.17$, point mutation rate $u = 10^{-9}$.
Colon: colonic adenocarcinoma; Rectal: rectal adenocarcinoma; Pancreas: pancreatic ductal adenocarcinoma.

from clinical experience. If there is even one possible mutation that can in principle confer resistance to both drugs, then our model shows that dual therapy has also only a small chance of curing patients, even those with the smallest tumor burden. In our cohort of 22 patients, none are expected to be cured under these circumstances (**Table 1**). Only if there are no potential mutations that can confer cross-resistance will dual therapy be successful in eradicating all lesions. In the cohort described in **Table 1**, we calculate that eight patients (those with the smallest tumor burden) would have >95% probability of cure. Those with the largest tumor burden would still have a >20% probability of tumor recurrence. Additional simulations show that therapy with three agents will also not cure patients if there is even one mutation that can confer resistance to all three agents. Similar conclusions hold if we vary parameter values within a reasonable range (**Supplementary file 3**). We note that in patients whose tumors have high cell turnover (time between cell divisions of 1 day, corresponding to $b = 1$), even dual therapy with no cross-resistance mutations would be expected to fail in 37% of patients described in **Table 1** (**Supplementary file 3**).

Graphical representations of the simulated responses of two patients with multiple metastatic lesions are shown in **Figure 4**. With monotherapy in patient N1 (**Figure 4A**), all lesions are predicted

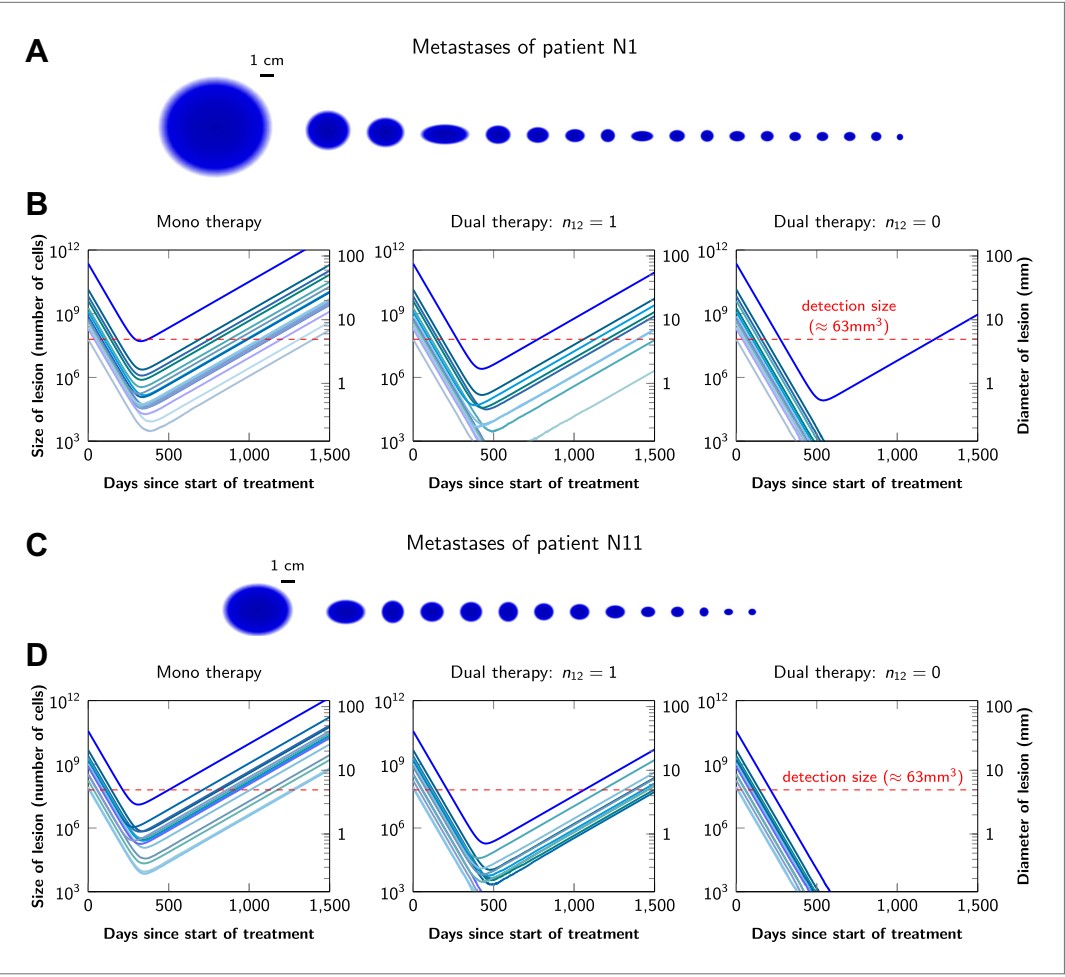

**Figure 4**. Treatment response dynamics to monotherapy and dual therapy in two patients. (**A**) Depiction of all 18 detectable metastases in patient N1, who had a particularly heavy tumor burden (scale 1:4). (**B**) Simulated treatment of patient N1, comparing monotherapy with $n = 50$ resistance mutations and dual therapy with $n_1 = n_2 = 50$ resistance mutations to the individual drugs and one ($n_{12} = 1$) or no ($n_{12} = 0$) cross-resistance mutations to both drugs. (**C**) Depiction of all 14 detectable metastases in patient N11, who had a more typical tumor burden (scale 1:4). (**D**) Simulated treatment of patient N11. Parameter values for simulations in (**B**) and (**D**): birth rate $b = 0.14$; death rate $d = 0.13$; death rate for sensitive cells during treatment $d' = 0.17$; point mutation rate $u = 10^{-9}$.

to regress, but then recur within a year or so after the initiation of therapy (*Figure 4B*, left panel). Treatment failure in most lesions would also occur after dual therapy when there is at least one mutation that could confer resistance to both agents, although the length of remission will be longer than with monotherapy (*Figure 4B*, middle panel). In patient N11, with less disease burden, dual therapy will fail to eradicate several of the lesions when there is a possibility of a single cross-resistance mutation, but there is hope of cure if no such cross-resistance mutations are possible (*Figure 4C,D*).

In current clinical practice, it is common to administer targeted agents sequentially: once relapse occurs, a second, often experimental, agent is administered. The model described above can also be used to predict the relative effectiveness of sequential vs simultaneous therapies of a single lesion with two drugs. When there is a possibility of a single mutation conferring resistance to both drugs, sequential combination therapy will 'always' fail. In ~74% of lesions, the failure is due to mutations that were present prior to the treatment with the first drug, whereas in ~26% of the lesions, failure is due to the development of cells resistant to drug 2 during treatment with drug 1 (*Figure 5A* and *Figure 5—figure supplement 1*). With simultaneous treatment, it is possible to eradicate ~26% of the lesions even when cross-resistance mutations are possible (*Figure 5B*). When there is no possibility of a mutation conferring cross-resistance to

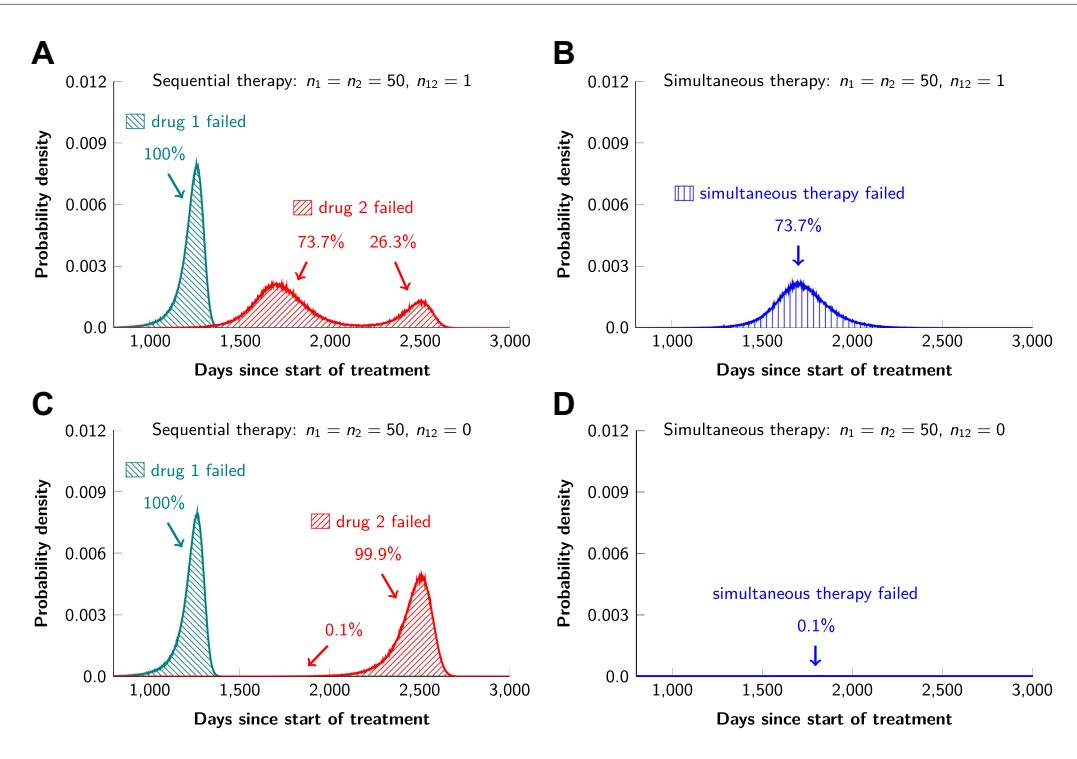

**Figure 5**. Sequential vs simultaneous therapy with two drugs. (**A**) If there is even a single mutation that confers cross-resistance to both drugs ($n_{12} = 1$), then sequential therapy will fail in all cases. In 73.7% of the cases, this failure is due to the exponential growth of fully resistant cells that were present at the start of treatment. In the remaining 26.3% of cases, the failure is due to resistance mutations that developed during therapy with the first drug. (**B**) With simultaneous therapy, 26.3% of patients can be cured under the same circumstances. In the remaining patients (73.7%), cross-resistant mutations existed prior to the therapy and their expansive growth will ensure treatment failure whether treatment is simultaneous or sequential (see **Figure 5—figure supplement 1** for further details). (**C**) and (**D**) If the two drugs have no resistance mutations in common ($n_{12} = 0$), then simultaneous therapy is successful with a probability of 99.9% while sequential therapy still fails in all cases.

The following figure supplements are available for figure 5:

**Figure supplement 1**. Examples for the evolution of resistance during sequential therapy.

both drugs, the differences are even more striking: sequential therapy fails in 100% of cases (*Figure 5C*), whereas simultaneous therapy succeeds in >99% of lesions of the identical size (*Figure 5D*).

One of the most important aspects of the cancer stem cell hypothesis revolves around therapeutic resistance. Evidence to date does not indicate that cancer stem cells are innately resistant to either single drugs or drug combinations. However, the precise proportion of cancer stem cells (among all cancer cells) has a dramatic effect on the development of resistance. This effect can be studied using our model if we use the number of cancer stem cells as an effective population size in our formulas and adjust other parameters to account for the stem cell dynamics (*Tomasetti and Levy, 2010*) (i.e., the birth rate should correspond to the rate of symmetric renewal, the rate of symmetric differentiation should be added to the death rate, and an effective mutation rate for stem cells should be introduced to account for mutations that occur during asymmetric division). For example, if cancer stem cells represent only 0.1% of cancer cells, then the development of resistance to single agents or combinations is roughly 0.1% as likely as if 100% of the cancer cells have the capacity to repopulate the tumor. The fraction of cancer stem cells appears to be this low in CML, perhaps explaining the remarkable success of imatinib (*Michor et al., 2005*). In solid tumors, however, the fraction of cancer stem cells seems much higher, usually higher than 5% and in some cases close to 100% (*Shackleton et al., 2009*). This issue is further complicated by the fact that the situation is plastic, with non-stem cells converting to cancer stem cells under certain conditions (*Gupta et al., 2009*). As better approaches to quantify

cancer stem cells in solid tumors become available, our estimates of the likelihood of therapeutic success will be improved.

If resistance has a fitness cost, then we expect a smaller number of resistant cells at the start of treatment and correspondingly a higher chance of treatment success. We used computer simulations to verify our results in the case when there is a cost for resistance, by assuming that each resistance mutation decreases the net growth rate of the cell by up to 10%. The results are shown in *Table 2*. For combination therapies with drugs that have resistance mutations in common, the probability of eradicating a lesion is only marginally affected by costly resistance. For dual therapies with no cross-resistance mutations, treatment has a high chance of eradicating all but the largest lesions, whether or not resistance is costly. In the case of large lesions with high cell turnover rates (the case in which even dual therapies with no cross-resistance might fail), costly resistance increases the chance of treatment success. For example, if each resistance mutation decreases the net growth rate of cells by 10%, the probability that dual therapy with no cross-resistance mutations will eradicate a lesion of size $10^{11}$ in which cells divide on average every day is 68% (compared with 47% in the case of neutral resistance).

Some therapies may directly eliminate tumor cells ($d' > d$), whereas others may impede their division ($b' < b$). Our formulas account for both of these possibilities. Overall, the rate $b' - d'$ of tumor decline is of primary importance, and whether this is achieved by eliminating cells or suppressing division has only a minor effect on treatment outcomes. For example, consider a dual therapy with $n_1 = n_2 = 50$, $n_{12} = 1$, applied to a lesion of size $M = 10^9$, with other parameters as inferred from our dataset. If this therapy shrinks the tumor at rate −0.03 per day by increasing cell death, the eradication probability is 26%. If the therapy instead suppresses division, this probability increases to 29%, because there are fewer chances for resistance mutations during treatment.

While our typical parameter values are derived from the melanoma dataset, our analytical results can accommodate parameter values from any other type of cancer, once they become available. Furthermore, our results are qualitatively robust across a wide range of birth and death rates (*Supplementary file 3*). The parameters with the strongest effects on the success of combination treatments—apart from the number of cross-resistance mutations—are lesion sizes and point mutation rate. Thus, we expect that combination treatments will be more effective in cancers with small fractions of tumor stem cells (small effective population size of lesions) and less effective in cancers with significantly increased point mutation rates.

## Discussion

Our conclusions are highly relevant for the expanding development and use of targeted agents for cancer therapy. Most importantly, they show that even if there is one genetic alteration within any of the 6.6 billion base pairs present in a human diploid cell that can confer resistance to two targeted agents, therapy with those agents will not result in sustained benefit for the majority of patients with advanced disease. The same result is obtained with triple therapy; if there is the possibility of a mutation

**Table 2.** Simulation results for the probability of treatment failure when resistance is costly

| Dual therapy: | | Number | | Probability of treatment failure | | | |
|---|---|---|---|---|---|---|---|
| $n_1 = n_2$ | $n_{12}$ | of cells | Birth rate | c = 0% | c = 1% | c = 5% | c = 10% |
| 50 | 0 | $10^9$ | 0.14 | 0.0 | 0.0 | 0.0 | 0.0 |
| 50 | 0 | $10^9$ | 1 | 0.01 | 0.01 | 0.01 | 0.0 |
| 50 | 1 | $10^9$ | 0.14 | 0.74 | 0.73 | 0.72 | 0.7 |
| 50 | 1 | $10^9$ | 1 | 0.74 | 0.74 | 0.72 | 0.7 |
| 50 | 0 | $10^{11}$ | 0.14 | 0.12 | 0.11 | 0.08 | 0.06 |
| 50 | 0 | $10^{11}$ | 1 | 0.53 | 0.51 | 0.42 | 0.32 |
| 50 | 1 | $10^{11}$ | 0.14 | 1.0 | 1.0 | 1.0 | 1.0 |
| 50 | 1 | $10^{11}$ | 1 | 1.0 | 1.0 | 1.0 | 1.0 |

Each resistance mutation reduces the net growth rate by a factor $c$ via a decrease of the birth rate $b$. Parameter values are death rate, $d = b - 0.01$, death rate for sensitive cells during treatment, $d' = b + 0.03$, point mutation rate, $u = 10^{-9}$. The simulation results are averages over $10^6$ runs per parameter combination.

conferring cross-resistance to three drugs, lesions of the size commonly observed in patients with advanced cancers will always recur. Similar conclusions were reached by *Komarova et al. (2009)*, who showed that a combination of three current targeted drugs for CML will not be beneficial over a combination of two such drugs due to cross-resistance. Our formulas could be used to develop an optimum in vitro assay to detect the existence of cross-resistance mutations for a given drug combination.

The development of drugs that act through distinct pathways will therefore be essential for the success of combination therapies in the clinic. Although this seems feasible in principle, there are a number of observations suggesting that it will be difficult in practice. For example, it has been shown that the increased expression of growth factors (such as hepatocyte growth factor) can confer resistance to a variety of drugs that inhibit kinases functioning through different pathways (*Straussman et al., 2012*; *Wilson et al., 2012*). Moreover, it is well known that mutations in several different genes, including those encoding ABC transporters, can confer resistance to many different drugs (*Lavi et al., 2012*). Drugs that have very different chemical structures, in addition to distinct mechanisms of action, may be required to circumvent these resistance mechanisms.

Our results are not readily applicable to therapies that rely on the immune destruction of tumors (*Kirkwood et al., 2012*), such as those employing CTLA-4 (*Hodi et al., 2010*), PD1 (*Topalian et al., 2012*), or CD19-CARs (*Grupp et al., 2013*). This promising line of therapy relies on an ongoing battle between cancer cells and the immune system. The immune system, unlike small molecule compounds, can replicate and evolve, and the factors underlying therapeutic success or failure are not sufficiently understood to allow useful modeling at this point. Once the mechanisms underlying the failures of immune modulators become more apparent, it will be important to try to understand why long-term control of disease is more common with these therapies than with small molecule drugs.

Our results on sequential vs simultaneous therapy with two or more agents (*Figure 5*) are in agreement with previous results (*Katouli and Komarova, 2011*) and have immediate practical implications even while new combinations are being developed. Sequential administration of targeted agents is often used to treat patients, for a variety of medical and economic reasons. Our data show that this sequential administration precludes any chance for cure—even when there are no possible mutations that can confer cross-resistance (*Figure 5C*). And when there are potential mutations conferring cross-resistance to two or more agents, simultaneous administration offers some hope for cure while there is no hope with sequential therapy (*Figure 5A*). The realization of the advantages of simultaneous vs sequential dual therapy will hopefully stimulate efforts to combine agents much earlier in the drug development process.

## Materials and methods

### Model

We model tumor growth and evolution as a continuous time multitype branching process (*Athreya and Ney, 1972*; *Goldie and Coldman, 1998*; *Komarova and Wodarz, 2005*). In the case of two drugs, there are four possible types: 00, 01, 10, and 11, where zeros indicate sensitivity to a drug and ones indicate resistance. For example, type 01 is sensitive to drug 1 and resistant to drug 2.

Our model includes two phases: pretreatment and treatment. The pretreatment phase is initiated with a single fully sensitive cell (type 00 for two drugs). During this phase, all cell types reproduce at rate $b$ and die at rate $d$. The offspring of a type 00 cell has probability $un_1$ of being type 10, $un_2$ of being type 01, $un_{12}$ of being type 11, and otherwise is of type 00. The offspring of a type 10 cell has probability $u(n_2 + n_{12})$ of being of type 11 and otherwise is of type 10; similar probabilities apply to type 01. Type 11 cells produce only type 11. These formulas generalize in straightforward manner to combination therapy with three or more drugs.

The pretreatment phase ends, and the treatment phase begins, when there are a total of $M$ cells. During the treatment phase, all cell types that are sensitive to one or more drugs have birth rate $b'$ and death rate $d'$; fully resistant cells maintain the pretreatment birth and death rates. Mutation probabilities are unchanged.

### Analysis

Our mathematical analysis of dual therapy is based in part on a recently discovered exact solution to the two-type branching process (*Antal and Krapivsky, 2011*). Detailed proofs of all results are provided in *Supplementary file 1*.

## Computer simulations

We use Monte Carlo computer simulations to confirm our analytical results and improve our understanding of the evolutionary dynamics during cancer treatment. The developed tool is an enhanced version of TTP (Tool for Tumor Progression) where the discrete time branching processes are replaced by continuous time branching processes (*Reiter et al., 2013*). Moreover, the new version also simulates tumor dynamics during treatment with several drugs. The simulations implement a multitype birth–death branching process using the specified parameter values. For cell subpopulations with less than $10^4$ cells, the process is simulated exactly; for larger subpopulations, a deterministic (exponential growth) approximation is used in the interest of efficiency. Within this deterministic approximation, the timing of appearances of new mutations is simulated using an adapted version of the Gillespie algorithm (*Gillespie, 1977*). Between $10^6$ and $10^8$ runs are used for each parameter combination.

To study the consequences of costly resistance, we suppose that each resistance mutation reduces the cell division rate such that the net growth rate is decreased by a factor $c$ representing the metabolic costs of resistance. For example, cells with two resistance mutations divide at rate $(b - d)(1 - c)^2 + d$.

# Additional information

## Funding

| Funder | Grant reference number | Author |
|---|---|---|
| Foundational Questions in Evolutionary Biology Fund | | Ivana Bozic, Benjamin Allen, Tibor Antal, Martin A Nowak |
| European Research Council Start Grant | 279307 | Johannes G Reiter, Krishnendu Chatterjee |
| FWF (The Austrian Science Fund) Grant | S11407-N23, P23499-N23 | Johannes G Reiter, Krishnendu Chatterjee |
| Microsoft Faculty Fellow Award | | Johannes G Reiter, Krishnendu Chatterjee |
| The John Templeton Foundation | | Martin A Nowak |
| The Danny Federici Melanoma Fund | | Paul B Chapman |
| John Figge Melanoma Fund | | Paul B Chapman |
| The Virginia and D. K. Ludwig Fund for Cancer Research | | Luis A Diaz Jr, Bert Vogelstein |
| National Cancer Institute | contract N01-CN-43309 | Luis A Diaz Jr, Bert Vogelstein |
| National Institutes of Health | CA129825, CA43460, CA57345 | Luis A Diaz Jr, Bert Vogelstein |
| National Colorectal Cancer Research Alliance | | Luis A Diaz Jr, Bert Vogelstein |

The funders had no role in study design, data collection and interpretation, or the decision to submit the work for publication.

## Author contributions

IB, JGR, BA, MAN, Designed the study, performed mathematical analysis and computer simulations, analyzed data and wrote the manuscript; TA, KC, Performed mathematical analysis and computer simulations, provided input to the manuscript; PS, YSM, AY, NK, DTL, EJL, PBC, LAD, Contributed data, analyzed data and provided input to the manuscript; BV, Designed the study, contributed data, analyzed data and wrote the manuscript

## Ethics

Human subjects: Work was reviewed by the institutional review board of Memorial-Sloan Kettering Cancer Center and was granted an exemption as per 45 CFR 46.101.b (4). A waiver for HIPAA Authorization and informed consent was granted as per 45 CFR 164.512(i)(2)(ii) and 45 CFR 46.116(d). The study protocols (ClinicalTrials.gov # NCT01459614 and Johns Hopkins Protocol # J0545 and J0746) were approved by the Johns Hopkins Institutional Review Boards and all patients signed a written consent form.

## Additional files

**Supplementary files**

• Supplementary file 1. Mathematical proofs.

• Supplementary file 2. Lesion sizes of patients who failed conventional treatments.

• Supplementary file 3. Probability of combination therapy failure in patients.

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
