## [Decision Letter]

Thank you for sending your work entitled “Evolutionary dynamics of cancer in response to targeted combination therapy” for consideration at *eLife*. Your article has been favorably evaluated by a Senior editor and 4 reviewers, one of whom is a member of our Board of Reviewing Editors.

The following individuals responsible for the peer review of your submission want to reveal their identity: Carl Bergstrom (Reviewing editor) and Rustom Antia (peer reviewer).

The Reviewing editor and the other reviewers discussed their comments before we reached this decision, and the Reviewing editor has assembled the following comments to help you prepare a revised submission.

The reviewers all found the paper to be an important contribution based on elegant modeling work. They raised the following points for consideration:

1) The authors state that the models explored in the context of previous studies are not applicable to describe solid tumors. It would help the reader if it were then explained in what way the new model differs from the old ones, and how its design helps to handle solid tumors. To this end, it would be helpful to discuss how the spatial structure of the tumor makes for non-homogenous growth and death rates and so on, and the limitations of the branching process model for dealing with these complexities.

2) The authors should consider moving supplementary figure 3 into the main text. One referee feels that this would greatly enhance the impact of the paper by making the result immediately and intuitively clear.

3) The clinical dataset used to generate the birth and death rates is from BRAF mutant melanoma. What is the evidence that the results can be extrapolated to all cancers?

4) The authors should comment on whether/how their modeling studies apply to the recent reports of long-term responses (cures?) with single agent immunotherapy (CTLA-4, PD1, CD19-CARs).

5) The authors make the assumption that the time between cell doublings is 7 days based on a single reference (Cell production rates in human tissues and tumours and their significance. Part II: clinical data. European journal of surgical oncology: the journal of the European Society of Surgical Oncology and the British Association of Surgical Oncology; doi: 10.1053/ejso.1999.0907). This is a difficult measurement to make and could be inaccurate. How would the predictions from the model change if this time were much different?

6) Near the end of the manuscript the authors make the point that the probability of cure with one or two drugs is greatly increased in tumors where the only cells that contribute to relapse are rare cancer stem cells. This is a logical and important distinction but seems to be quickly dismissed as not relevant clinically, with the possible exception of CML. Since the cancer stem cell question remains murky, this deserves more comment.

7) All of the calculations are based on tumor cell number (for obvious reasons) but tumor size (on CT scan) is used by clinicians. We encourage the authors to include both measurements on relevant figures, perhaps in the form of a conversion scale, to make it easier for clinicians to appreciate the clinical relevance of the modeling predictions.

8) The authors may wish to consider the presentation of the mathematical results, which currently are spread among several different parts of the manuscript and supplementary information. Given that *eLife* doesn't impose constraints on length, we wonder whether it would be possible to better structure the manuscript so that the reader does not have to flip between so many different sections to follow the story. This is not to say that the separation is without merit–the main story is well told with the mathematical details set elsewhere. But it seems worth at least considering whether the supplementary information could be consolidated.

---

## [Author Response]

*1) The authors state that the models explored in the context of previous studies are not applicable to describe solid tumors. It would help the reader if it were then explained in what way the new model differs from the old ones, and how its design helps to handle solid tumors. To this end, it would be helpful to discuss how the spatial structure of the tumor makes for non-homogenous growth and death rates and so on, and the limitations of the branching process model for dealing with these complexities*.

We were referring to previous clinical studies that report the success of chemotherapeutic agents in leukemias and combination treatments in HIV, not to theoretical models that predict the success of these therapies. Our point is simply that the success of combination-targeted therapies in solid tumors cannot be inferred from successes in treating leukemias. Our primary advances with regard to previous branching-process models are in the development and incorporation of new mathematical techniques yielding exact closed-form results, and our use of new clinical data to obtain parameter values applicable to solid tumors. We clarified the explanation on how our model differs from previous ones and noted the limitations of our model to deal with non-homogeneous growth.

*2) The authors should consider moving supplementary figure 3 into the main text. One referee feels that this would greatly enhance the impact of the paper by making the result immediately and intuitively clear*.

We agree with the referees and we have now made this Figure 3 within the main text. The figure compares the formula for the probability that treatment will eradicate a lesion ([Disp-formula equ4]) with simulation results, so we refer to it when discussing that formula.

*3) The clinical dataset used to generate the birth and death rates is from BRAF mutant melanoma. What is the evidence that the results can be extrapolated to all cancers*?

While the typical parameter values we use are from the melanoma dataset, our analytical results can accommodate parameter values from any other type of cancer, once they become available. Furthermore, our results are qualitatively robust across a wide range of birth and death rates (Supplementary File 3). The parameters with the strongest effect on the success of combination treatment–apart from the number of cross-resistance mutations–are lesion size and point mutation rate. Thus we expect combination treatment to be more effective in cancers with small fractions of tumor stem cells (resulting in a small effective population size) and less effective in cancers with increased point mutation rates.

*4) The authors should comment on whether/how their modeling studies apply to the recent reports of long-term responses (cures?) with single agent immunotherapy (CTLA-4, PD1, CD19-CARs)*.

Our results are not readily applicable to therapies that rely on the immune destruction of tumors, because immune cells can replicate and evolve, unlike small molecule compounds. We have added a discussion pertaining to this issue.

*5) The authors make the assumption that the time between cell doublings is 7 days based on a single reference (Cell production rates in human tissues and tumours and their significance. Part II: clinical data. European journal of surgical oncology: the journal of the European Society of Surgical Oncology and the British Association of Surgical Oncology; doi: 10.1053/ejso.1999.0907). This is a difficult measurement to make and could be inaccurate. How would the predictions from the model change if this time were much different*?

As we mention in the main text, we use doubling time of 7 days as a typical value, but find that our conclusions hold for doubling times ranging from 1 to 14 days as well ([Supplementary-material SD4-data]).

*6) Near the end of the manuscript the authors make the point that the probability of cure with one or two drugs is greatly increased in tumors where the only cells that contribute to relapse are rare cancer stem cells. This is a logical and important distinction but seems to be quickly dismissed as not relevant clinically, with the possible exception of CML. Since the cancer stem cell question remains murky, this deserves more comment*.

We agree with the reviewers that the issue of cancer stem cells has important clinical implications for the success of single and combination therapies. We have added a discussion of these points to the manuscript.

*7) All of the calculations are based on tumor cell number (for obvious reasons) but tumor size (on CT scan) is used by clinicians. We encourage the authors to include both measurements on relevant figures, perhaps in the form of a conversion scale, to make it easier for clinicians to appreciate the clinical relevance of the modeling predictions*.

We have now added tumor diameter (in addition to cell number) to all relevant figures.

*8) The authors may wish to consider the presentation of the mathematical results, which currently are spread among several different parts of the manuscript and supplementary information. Given that eLife doesn't impose constraints on length, we wonder whether it would be possible to better structure the manuscript so that the reader does not have to flip between so many different sections to follow the story. This is not to say that the separation is without merit–the main story is well told with the mathematical details set elsewhere. But it seems worth at least considering whether the supplementary information could be consolidated*.

To simplify the presentation of the mathematical results, as suggested by the reviewers, we have moved the mathematical results from the Methods section to the Results section in the main text. We have also consolidated the supplementary information into a more linear structure.